# Immune Regulation and Immune Therapy in Melanoma: Review with Emphasis on CD155 Signalling

**DOI:** 10.3390/cancers16111950

**Published:** 2024-05-21

**Authors:** Li-Ying Wu, Su-Ho Park, Haakan Jakobsson, Mark Shackleton, Andreas Möller

**Affiliations:** 1School of Biomedical Sciences, Faculty of Health, Queensland University of Technology, Brisbane, QLD 4059, Australia; n10741046@qut.edu.au; 2JC STEM Lab, Department of Otorhinolaryngology, Chinese University of Hong Kong, Shatin, Hong Kong SAR, China; su-ho.park@cuhk.edu.hk; 3Li Ka Shing Institute of Health Sciences, Chinese University of Hong Kong, Hong Kong SAR, China; 4Department of Medical Oncology, Paula Fox Melanoma and Cancer Centre, Alfred Health, Melbourne, VIC 3004, Australia; h.jakobsson@alfred.org.au; 5School of Translational Medicine, Monash University, Melbourne, VIC 3004, Australia

**Keywords:** melanoma, immunotherapy, immune regulation, CD155, tumour microenvironment

## Abstract

**Simple Summary:**

For melanoma patients, the most promising curative treatment is immunotherapy. Here, we summarise current immunotherapy strategies and their indications and challenges, and provide future directions of immunotherapy development. Several therapy resistance mechanisms have been described, resulting in primary refractory disease or resistance development in patients. Our increasing knowledge of immune regulation within the tumour microenvironment identifies potential alternate immunotherapy targets, including CD155. CD155 and its receptor, TIGIT, have been shown to be highly expressed in therapy-resistant melanoma cells and interference with both is therefore considered a possible immunotherapeutic strategy. This review describes the immune regulation within the melanoma tumour microenvironment, how and why immunotherapies work, and why CD155 might be an ideal target for the next generation of anti-melanoma immunotherapies.

**Abstract:**

Melanoma is commonly diagnosed in a younger population than most other solid malignancies and, in Australia and most of the world, is the leading cause of skin-cancer-related death. Melanoma is a cancer type with high immunogenicity; thus, immunotherapies are used as first-line treatment for advanced melanoma patients. Although immunotherapies are working well, not all the patients are benefitting from them. A lack of a comprehensive understanding of immune regulation in the melanoma tumour microenvironment is a major challenge of patient stratification. Overexpression of CD155 has been reported as a key factor in melanoma immune regulation for the development of therapy resistance. A more thorough understanding of the actions of current immunotherapy strategies, their effects on immune cell subsets, and the roles that CD155 plays are essential for a rational design of novel targets of anti-cancer immunotherapies. In this review, we comprehensively discuss current anti-melanoma immunotherapy strategies and the immune response contribution of different cell lineages, including tumour endothelial cells, myeloid-derived suppressor cells, cytotoxic T cells, cancer-associated fibroblast, and nature killer cells. Finally, we explore the impact of CD155 and its receptors DNAM-1, TIGIT, and CD96 on immune cells, especially in the context of the melanoma tumour microenvironment and anti-cancer immunotherapies.

## 1. Introduction

Skin cancer is divided into melanoma and non-melanoma skin cancer, with the latter including basal cell carcinoma and squamous cell carcinoma [1]. While non-melanoma skin cancers have very high incidences, their mortalities are comparably small, whereas melanoma is a major cause of cancer-related death [2], with Australia having the highest melanoma incident worldwide [3,4]. Although melanoma incidence can be significantly reduced by lifestyle interventions such as regular use of sunscreen [5], nevertheless, there is a significant proportion of the population, especially in the 20 to 39 age range, diagnosed every year [4].

Melanocytes are the cells that produce melanin to absorb ultraviolet radiation (UV), and thus provide photoprotection and thermoregulation to the skin [6]. Overexposure to UV is considered the main risk factor for causing malignant transformation of melanocytes [7], which progresses into cutaneous melanoma [6,8]. Early-stage melanoma can be surgically removed, commonly resulting in good outcomes [2], with a 5-year survival rate of stage I melanoma patients of around 97–99% [9]. In contrast, the 5-year survival rate of advanced, metastatic stage IV patients is only 15–20% [9]. Melanoma is associated with the highest rate of genetic mutations amongst cancer types [10] and intertumoural heterogeneity [11], creating challenges for developing advanced melanoma treatments [10]. Common mutations in melanoma are BRAF and NRAS [8,12], but, in addition, approximately 70% of mutations in melanoma involve the mitogen-activated protein kinase (MAPK) pathway [13]. Thus, the BRAF inhibitor vemurafenib is a commonly used targeted therapy and improves melanoma survival rates [14]. However, not all patients with BRAF mutations benefited from the treatment, with some of them displaying initial or acquired drug resistance [14].

Another feature that makes melanoma so aggressive, but has recently been exploited as a therapeutic opportunity, is its high antigenicity [15,16] and immunogenicity [17,18]. Melanoma escapes immune surveillance by manipulating immune checkpoints, such as programmed cell death protein 1 (PD-1) [19], programmed death-ligand 1 (PD-L1), and cytotoxic T-lymphocyte-associated protein-4 (CTLA4) [20]. One of the recent therapeutic breakthroughs came from the inhibition of these immune checkpoints, which greatly improved advanced melanoma patient survival rates; however, not all the patients obtained a sustained clinical response [18,21,22,23,24], with some patients initially responding but developing immunotherapy resistance [25,26]. It has therefore become a focus to comprehensively understand the details of immune regulation in melanoma in order to provide new potential therapeutic targets to improve patient outcomes.

CD155, also known as poliovirus receptor (PVR), maintains the normal function of immune cells by controlling activating and inhibitory signals via interactions with DNAM-1, CD96, and TIGIT receptors [27]. CD155 is highly expressed in most melanoma cell lines [28] and involved in drug resistance to immune therapy in patients [29,30,31,32], and therefore has been suggested to be a potential target for co-inhibitory immune therapies [33].

Here, we will first discuss clinical outcomes of immune therapy in melanoma patients and the problems that need to be addressed to improve current treatment options. Additionally, we will comprehensively review and discuss immune regulation within the melanoma tumour microenvironment, including specific immune responses triggered in different cell lineages. The specific contributions of these lineages to melanoma immunosurveillance escape will be described, as well as their relative contributions to promoting tumour growth through regulating immune checkpoint proteins. We will also review currently used biomarkers shown to be predicting patient outcomes treated with immune therapies. Finally, we will evaluate how CD155 manipulates immune cells to inhibit anti-tumour immune response, its role in melanoma escape from immune surveillance, and how this knowledge could assist in the clinical management of melanoma patients.

## 2. Current Melanoma Patient Immunotherapy Strategies

The use of Coley’s toxin over 100 years ago to induce anti-tumour responses provided the inspiration for generations of cancer and immunology researchers to harness the power of the immune system in cancer therapy [34]. Immunotherapy for the treatment of cancer gained momentum in the late 1950s with the discovery by Issacs and Linderman [35] of interferon (IFN), which displayed anti-cancer efficacy, particularly against melanoma (Figure 1). This eventually led to the FDA approval of INF-a2b, which improved relapse-free survival in patients with fully resected melanoma [36]. In parallel, the use of high-dose interleukin (IL)-2 was found to induce objective responses in patients with metastatic melanoma, including complete and durable responses [37], and re-infusion of ex vivo processed and expanded tumour-infiltrating lymphocytes conferred impressive clinical benefit in small studies [38].

While these successes were encouraging, more reliably effective treatments were needed. A better understanding of interactions between the immune system and cancer, particularly mechanisms that inhibit the expansion and activity of cytotoxic T cells and that regulate evasion by cancer cells from immune attack [39], led to the development of more robust treatment strategies. The anti-tumour immune response in the tumour microenvironment is often dysfunctional. Tumour cells respond to immune activation through several mechanisms, including the expression of immunosuppressive moieties like PD-L1, which impairs T-cell responses [40]. T-cell priming may also be inadequate at the point of initial tumour antigen presentation, resulting in impaired initiation of anti-tumour immune responses through inhibitory molecules such as CTLA-4 [41,42].

The first approved therapy in targeting such immune resistance was ipilimumab, which inhibits CTLA-4 and increases generic and tumour-specific T-cell activation [43]. In a paradigm-changing trial in patients with metastatic or unresectable melanoma, ipilimumab demonstrated an improvement in median survival of 10 months compared to 6.4 months for gp100, a peptide vaccine [44]. This resulted in FDA approval of ipilimumab in 2011. While this was a significant improvement over existing standards of care, the response rate to single-agent ipilimumab was low at only 10.9%.

In 2014, the anti-PD-1 inhibitors pembrolizumab and nivolumab were approved for metastatic melanoma. The KEYNOTE-006 trial demonstrated improved overall survival of 32.7 months for pembrolizumab compared to the historical 15.9 months for ipilimumab [45]. Additionally, response rates to pembrolizumab were 33%, higher than those for ipilimumab, offering clinical meaningful improvements in disease-related symptoms [46] (Figure 1).

A few years later, the landmark Checkmate 067 study of ipilimumab and nivolumab combination therapy compared to nivolumab or ipilimumab alone set a new standard of care for metastatic melanoma [23]. This study demonstrated impressive and durable survival outcomes, with a median overall survival of 72.1 months in the combination arm compared to 39.6 months and 19.9 months for the nivolumab and ipilimumab arms, respectively. Overall survival was 49% at 6.5 years in the combination arm compared to 42% and 23% in the nivolumab and ipilimumab arms, respectively. Overall response rate in the combination was 58%, a significant improvement over single-agent therapy. Similarly to interferon and interleukin therapies, the major downside of this approach was the high levels of toxicity, with nearly all patients experiencing at least minor toxicity and rates of severe toxicity approaching 60%, nearly three times higher than single-agent anti-PD1 therapy in the same trial.

More recent developments have led to efficacious but less toxic immunotherapy combinations in melanoma. Relatlimab, an LAG-3 inhibitor, in combination with nivolumab, demonstrated a superior progression-free survival of 10.1 vs. 4.6 months in the RELATIVITY-047 study compared to nivolumab alone [47]. The toxicity of the relatlimab plus nivolumab combination was substantially lower than that seen in CheckMate 067, with severe toxicity seen in only 18.9% of patients.

The use of immunotherapy has also extended to the treatment of patients with locoregionally advanced disease who undergo curative-intent surgery, nowadays offered in both adjuvant [48,49] and neoadjuvant [50] treatment contexts. Interestingly, while the key studies that tested anti-PD1 in these contexts each showed substantial improvements in relapse-free survival, none of them has to date demonstrated an overall survival benefit compared to placebo treatment. This has raised the possibility that the patients with early-stage disease who benefit from adjuvant anti-PD1 therapy are the same patients who enjoy sustained remission after treatment of their advanced disease. Nevertheless, the strikingly improved relapse-free survival observed in the SWOG 1801 study of the pre-operative neoadjuvant followed by post-operative adjuvant pembrolizumab, compared to adjuvant pembrolizumab alone, suggests that surgery might induce adjuvant immunotherapy resistance in some patients, such that neoadjuvant therapy might eradicate disease in some patients who otherwise would not have responded in a sustained manner had their disease relapsed and metastasised widely [50].

Challenges and opportunities remain, however, in melanoma treatment. Between 40 and 65% of patients treated with immunotherapy may be primarily refractory to this treatment [51], leading to the need for alternative strategies. Later-line treatments for melanoma can be of limited benefit for patients without activating class 1 BRAF mutations, with objective response rates to single-agent ipilimumab following single-agent anti-PD1 therapy as low as 9%, although combination therapy with ipilimumab and nivolumab in this context is more promising, offering response rates of 28% [52]. In another approach, addition of the multi-kinase inhibitor lenvatinib to pembrolizumab demonstrated objective response rates of 21.4% in the phase II LEAP-004 trial, with a median overall survival of 14 months [53].

Immunotherapy is likely to remain a backbone of treatment for metastatic or unresectable melanoma, although many patients do not obtain the deep and durable responses that others enjoy. This highlights the urgent need for ongoing innovation in melanoma treatment through the understanding of primary and secondary mechanisms of resistance to immunotherapy, biomarkers to indicate those early, and strategies to overcome or avert these [54].

## 3. Immunoregulation Mechanisms within the Melanoma Tumour Microenvironment

Within the melanoma tumour microenvironment, the most abundant non-malignant cell populations are immune and endothelial cells as well as cancer-associated fibroblasts (CAFs) [55,56]. The expression of immune checkpoint proteins has been observed in a variety of cell types, including tumour endothelial cells (TECs) and myeloid-derived suppressor cells (MDSCs), which are involved in the regulation of immune suppressive signalling [57,58]. Understanding the immune suppressive regulation mechanisms in different cell types might identify potential targets for melanoma therapy (Figure 2).

### 3.1. Tumour Endothelial Cells (TECs)

Endothelial cells in the tumour microenvironment are essential mediators of neo-angiogenesis, providing the cancer cells with sufficient oxygen and nutrients, thus enabling tumour growth [73,74]. TECs and the resulting cancer vascular system have different phenotypes to normal, physiological endothelial cells and vessels [75,76]. Not only is the cancer vascular system leakier [75], but it also has a different structural morphology, with vessels made mostly of TECs associated with a limited number of pericytes, arranged in a more random fashion, and monocyte-derived cells, phenocopying endothelial cells [75]. TECs in particular trigger immune suppressive signals, resulting in CD8^+^ T-cell dysfunction [57], which can be overcome by blocking PD-L1, PD-1, or CTLA4 [57,77]. The expression of PD-L1 by TECs induces immune-suppressive CD4^+^ regulatory T cells (Tregs), which, in turn, upregulate IL-10 and TGF-β expression, resulting in reduced CD8^+^ T-cell proliferation and pro-inflammatory cytokine production, such as IL-2, TNFα, and IFN*γ* [57]. Dysfunctional CD8^+^ T cells then further promote tumour growth [57,77]. However, stimulator of IFN genes (STINGs)-induced IFN-β production by VEGFR2^+^ CD31^+^ TECs [77] or inhibition of PD-1 [77] or PD-L1 [57] reverses CD8^+^ T-cell dysfunction and promotes anti-tumour immune responses. Moreover, melanoma cells express adherent molecules, such as N-cadherin, integrin α4β1, and ALCAM, which allow melanoma cells to attach to TECs, further promoting invasion and metastasis [16,64].

### 3.2. Cancer-Associated Fibroblast (CAF)

Under physiological conditions, the activation of fibroblasts occurs in the process of wound healing and decreases once wound healing is completed [78]. Sustained activation of fibroblasts is observed under pathological conditions, such as cancer and chronic fibrosis [79]. These activated fibroblasts, also termed myofibroblasts, are spindle-shaped cells and highly express α-smooth muscle actin (α-SMA) and fibroblast activation protein (FAP) [78]. Myofibroblast activation has been suggested to enhance contraction and promote extracellular matrix (ECM) remodelling, which results in fibrosis [78]. Similar features of remodelling the ECM can be found in cancer-derived fibroblasts (CAFs) [80], which trigger mechanotransduction signalling pathways, such as the hippo pathway, to promote cancer cell proliferation [59,60] and epithelial–mesenchymal transition (EMT) [61,62]. In addition, the secretome of CAFs has been shown to promote tumour growth [63] and induce drug resistance to therapeutic BRAF inhibition [81]. CAF-secreted CXCL5 causes PI3K/Akt pathway activation in cancer cells, resulting in PD-L1 expression, which can be blocked by the inhibition of the CXCL5 receptor CXCR2 [63]. Together, these data suggest that CAFs are suitable therapeutic target cell lineages in melanoma [82].

### 3.3. Myeloid-Derived Suppressor Cells (MDSCs)

Under normal conditions, myeloid cells are critical innate immune cells, involved in activating adaptive immunity in response to pathogen-associated molecular signals, which trigger the differentiation of myeloid progenitors to mature monocytes and granulocytes to protect the host [83]. In the tumour microenvironment, tumour-associated factors such as VEGF, TGFβ, IL-6, and PD-1 impair myeloid cell functions, resulting in inhibition of the anti-tumour immune response, which further promotes tumour growth [84,85]. Downregulation of granulocyte macrophage progenitor differentiation increases PD-1 expression on myeloid cells, which suppress CD8^+^ T-cell activity and promote tumour growth [58]. This effect can be reversed by specifically targeting PD-1 on myeloid cells [58]. Although the inhibition of PD-1 restores CD8^+^ T-cell activity and enhances anti-tumour immune regulation, drug resistance has been found in advanced melanoma patients [26,86]. Multiple pathways drive resistance to PD-1 blockade by promoting MDSCs infiltration into the tumour microenvironment [25,26,65], which is one of the major causes for the inhibition of the anti-tumour immune response [26,85,87]. Tumour-derived exosomes released by various tumour cells induce the expansion and immunosuppression of MDSCs through exosomal proteins, such as PGE2, TGFβ, Hsp72, IL-10, or IL-16 [88,89]. Moreover, high MDSC abundance is correlated with poor treatment responses to the inhibition of immune checkpoint therapy in patients with advanced melanoma [86,90]. MDSCs are mainly divided into two sub-populations, monocytic-MDSCs (M-MDSCs) and polymorphonuclear-MDSCs (PMN-MDSCs) [91,92]. Both M-MDSCs [85] and PMN-MDSCs [26] are involved in immune-suppressive regulation in melanoma [87]. M-MDSCs are abundant in tumour tissues, suppressing both an antigen-specific and non-specific T-cell response, mainly relying on arginase 1 (Arg1), nitric oxide (NO), and immunosuppressive cytokines. PMN-MDSCs predominately locate to the peripheral lymphoid organs and activate antigen-specific suppression, and their function depends largely on reactive oxygen species (ROS) and peroxynitrite (PNT) [93]. M-MDSCs promote immune escape by the inhibition of CD8^+^ T-cell infiltration into tumour tissues, depending on CCR2-mediated signalling [94] or suppression of CD8^+^ T-cell proliferation, including using CCR5-related pathway signalling [95]. M-MDSCs have more potent suppressive effects on antigen-specific CD8^+^ T-cell proliferation than PMN-MDSCs, but PMN-MDSCs are the major immunosuppressor in cancer patients [96,97]. PMN-MDSCs accumulate in tumour tissues and represent a key adaptive resistance pathway to PD-1 blockade after CD8^+^ T-cell activation in CXCR2/CXCL5-dependent manners [26]. Several inflammatory cytokines, including CCL2, CCL5, and CXCL5, have been found to promote MDSCs infiltration, and correlate with a poor treatment outcome [26,98,99,100]. Furthermore, high CXCL cytokine family expression is also found in patients with different cancer types [101,102]. In a BRAF^V600E^/PTEN^−/−^ melanoma model, the inhibition of PD-1 increases CD8^+^ T-cell activity and IFN-*γ* secretion, which, in turn, increases Wnt5a expression in melanoma cells. Activation of Wnt5a triggers CXCL5 secretion through a YAP-dependent pathway, subsequently promoting PMN-MDSCs infiltration [26]. However, anti-PD-1 therapy using pembrolizumab decreases the level of PMN-MDSC (CD11b^+^CD33^+^HLA-DR^lo/−^CD15^+^) in melanoma patient PBMCs [103]. In melanoma-bearing mice, blocking CCR5/CCL5 interaction by mCCR5-Ig [104] reduces the migration and immunosuppressive potential of PMN-MDSCs (CD11b^+^Ly6G^+^Ly6C^lo^), resulting in increased survival [95]. IL-33 also has a role in the anti-tumour response to melanoma by not only increasing the secretion of INFγ and granzyme B from CD8^+^ T and NK cells as well as proliferation of CD4^+^ T cells [105,106] but also reducing the differentiation of PMN-MDSCs from bone marrow cells [106]. The capacity of MDSCs to induce Tregs is in turn inhibited by IL-33 treatment [106].

### 3.4. Cytotoxic T Cells (CD8^+^ T Cells)

When naïve CD8^+^ T cells recognise antigens on antigen-presenting cells (APCs), such as dendritic cells or macrophages in secondary lymphoid organs, they are activated and then differentiate into cytotoxic effector CD8^+^ T cells to control the antigens or pathogens [107]. Once antigens or pathogens are cleared, cytotoxic effector T cells undergo either of two fates: either apoptosis or differentiation into central memory T cells (T_CMs_) and effector memory T cells (T_EMs_) for a rapid response against subsequent infection [107]. T_EMs_, which migrate between blood and peripheral organs, have more immediate effects on immune surveillance than T_CMs_, but are not sustained longer term. In contrast, T_CMs_ sustain a more durable response by high rates of proliferation through IL-2 production in secondary lymphoid organs [108,109].

Upon tumour initiation, tumour antigens trigger cytotoxic CD8^+^ T-cell activation, which eliminates tumour cells by two main pathways: death ligands and granule exocytosis. Various death ligands, such as tumour necrosis factor (TNF)-α, Fas ligand, and TNF-related apoptosis-inducing ligand (TRAIL), are expressed on the membrane of CD8^+^ T cells or secreted from these cells and induce tumour cell apoptosis [110]. Granzymes (Gzms) are released from activated cytotoxic CD8^+^ T cells together with perforin, which induces pore formation in the plasma membrane of target cells to deliver Gzms into them, including into tumour cells [111,112]. Granzyme B (GzmB), the most potent pro-apoptotic serine protease, is commonly found in the granules of cytotoxic CD8^+^ T cells [113]. Anti-tumour responses are enhanced by IFN-γ secreted from activated CD8^+^ T cells through an increase in the expression of MHC class I antigens in tumour cells, thereby making these cancer cells a better target for CD8^+^ T cells [114]. In tumour tissues, the abundance of GzmB, perforin, TNF-α, and IFN-γ is significantly decreased in CD4^+^, CD8^+^ T cells, and NK cells compared to non-tumour tissues [115].

The tumour environment is characterised by a persistent antigen stimulation as well as chronic inflammatory conditions, which result in T cells progressing into a state of “dysfunction” or “exhaustion” [69,70]. Dysfunctional (also commonly named as exhausted) CD8^+^ T cells lose part of their function to kill tumour cells and appear in large amounts in tumour tissues alongside cytotoxic CD8^+^ T cells [69,71]. Especially in melanoma, dysfunctional CD8^+^ T-cell populations occur in high abundances of 5–80% of total tumour-infiltrating lymphocytes (TILs) [71]. Dysfunctional CD8^+^ TILs are not connected to effector cytotoxic CD8^+^ T cells, as minimal overlapping transcripts and TCR repertoires between cytotoxic and dysfunctional CD8^+^ T-cell pools exist [116]. Dysfunctional CD8^+^ T cells are characterised by expressing different inhibitory molecules, such as *LAG3*, *PDCD1* (encoding PD-1), *CXCL13*, *HAVCR2* (encoding Tim3), *ENTPD1* (encoding CD39), *CTLA4*, and *TIGIT*, which are all less expressed in memory-like T cells in melanoma [71,72]. However, dysfunctional CD8^+^ T cells still have the capacity of tumour antigen recognition, discovered in autologous single-cell melanoma digests [70]. This cell compartment highly expresses CD39 (*ENTPD1*) and CD103 (*ITGAE*), whereas cytotoxic effector KLRG1 is absent [70]. CD39/CD103-double-positive CD8^+^ TILs are found in high abundances in cancer tissues isolated from patients with melanoma, head and neck squamous cell carcinoma, lung, ovarian, and rectal cancer. Both CD39 and CD103 are upregulated by TGF-β and TCR stimulation, suggesting that CD39^+^/CD103^+^ CD8^+^ TILs are enriched in the tumour microenvironment [117]. PD-1 is highly co-expressed with CD103, one of the tissue-resident markers on intra-epithelial CD8^+^ TIL in human ovarian cancer and non-small-cell lung cancer [118,119]. PD-1 expression relies on the persistence of antigen exposure to CD8^+^ T cells by its TCR stimulation [117,118]. PD-1 negatively regulates the activation of the co-stimulatory molecule CD28 by dephosphorylation in T cells after interaction with PD-L1 [120], and PD-1^+^Tim3^+^CD8^+^ TILs lose CD28 expression [121]. The blockade of the immune checkpoint PD-1 by anti-PD-1/PD-L1 induces the clinical benefit of tumour patients [122,123]. Anti-PD-1 treatment increases the transfer of Tcf1^+^Tim3- progenitor exhausted (dysfunctional) CD8^+^ T cells into the B16 melanoma tumour, but not of their Tcf1-Tim3^+^ terminally exhausted counter parts [124]. In addition, PD-1/PD-L1 blockade induces the selective expansion of PD-1^Int^CD44^Hi^ exhausted CD8^+^ T cells in LCMV chronic infection [125]. Anti-PD-L1 therapy is sensitive to the proliferation of CD28-positive cells among CD8^+^ T cells in lung cancer [121].

TIGIT, which is one of the CD155 receptors, has been found to be highly expressed in exhausted T cells [126]. After binding to CD155, TIGIT downregulates DNAM-1 expression, which prevents cytotoxic T cells from transforming into exhausted T cells, to promote cytotoxic T-cell dysfunction and suppress cytotoxic T-cell proliferation (Figure 2). TIGIT^+^CD8^+^ T cells are highly expressed in PBMCs from patients with cervical cancer [126], but no differences have been found in PBMCs from melanoma patients compared with healthy donors [127]. However, TIGIT-positive populations are high among NY-ESO-1 (TA; tumour antigen)-specific CD8^+^ T cells from PMBCs [127]. In addition, TIGIT expression is higher in CD8^+^ TILs than circulating CD8^+^ T cells in PBMCs in melanoma patients [127]. TIGIT downregulates the expression of TNF-α and IFN-γ in CD8^+^ T cells [126,127]. Especially, high numbers of infiltrating TIGIT^+^CD8^+^ T cells into the tumour diminish the anti-tumour immunity, associated with a low expression of TNF-α, IFN-γ, and GzmB [128].

### 3.5. Natural Killer Cells (NK Cells)

Natural killer cells (NK cells) are innate lymphoid cells that can directly kill pathogens or target cells without requiring triggers by other innate immune cells [129]. The appearance of CD56 on human NK cell surfaces indicates the differentiation from immature NK (iNK) cells to mature NK (mNK) cells [130]. CD56-negative iNK cells mostly differentiate into CD56^dim^-mNK cells that are converted through CD56^bright^-mNK cells, or directly [130,131,132,133]. The maturation of NK cells is also tightly associated with CD16 expression [131,133,134]. CD56^bright^ mNK cells, which have a high inflammatory cytokine secretion capacity, are mainly found in lymphoid tissues [133,135,136]. CD56^dim^CD16^+^-mNK cells, which are mostly found in peripheral blood, have higher cytolytic activity [131,135,137]. Similar subsets of NK cells were found in mice, with immature NK cells being associated with CD27^+^CD11b^−^ phenotypes and terminally mature NK cells with CD27^−^CD11b^+^KLRG1^+^ [130,134]. The functionally activated CD107a^+^ NK cells use perforin and GzmB as granule-mediated cytotoxicity against target tumour cells as an early stage of effect, while NK cells subsequently switch to CD95L death-receptor-mediated cytotoxicity as a final stage of cell killing [66]. CD107a protects against the perforin-mediated self-destruction in NK cells by reducing perforin binding during target cell lysis [138]. The expression of CD107a is higher on CD56^dim^CD16^−^NK cells than CD56^dim^CD16^bright^-NK cells [139], supporting that CD107a is supposed to decrease alongside CD16 in CD56^dim^-NK cells after targeting tumour cells. Upon pathogen infection or tumour formation, NK cells migrate to these sides and release cytokines to enhance the functional killing effect against target cells [140]. Cytokines, such as IL-15 and IL-18, have been shown to be essential for NK cell maturation and function [140]. IL-15 promotes NK cell maturation and enhances NK cell cytokine production [67,141]. IL-18 facilitate NK cell cytokine production and cytolysis [142,143]. Moreover, IL-15 and IL-18 have been found to promote NK cell activation and their undergoing of memory-like differentiation in melanoma [67]. IL-12 enhances the proliferation of NK cells because this cytokine binds to the α chain of the IL-2 receptor and induces CD25 [140]. IL-12 also promotes IFN-γ production and IL-2, IL-15, and IL-18 signalling in a positive loop with each IL receptor (i.e., IL-2R, IL-12R, IL-15R, and IL-18R) in NK cells [144]. Memory-like NK cells increase IFN-*γ* production to kill melanoma cells in an NKG2D- and NKp46-dependent manner [67].

Within the immune cell population, there are only small amounts of NK cells infiltrating into the tumour microenvironment [145]. Despite the low abundance, NK cell infiltration has been shown to significantly affect cancer prognosis [146,147]. Protein signatures of tumour-infiltrating NK cells are also quite different from those of circulating blood NK cells in melanoma patients [147,148]. Some cytokines, including XCL1 and XCL2, were found to be highly expressed specifically in tumour-infiltrating NK cells [148]. NK cells secrete cytokines, such as XCL1 [149], CCL5 [149] and FLT3LG [149], to stimulate dendritic cell (DC) infiltration into the tumour microenvironment and interact with cytotoxic CD8^+^ T cells to improve patient responses to immunotherapy [149,150,151]. Thus, the infiltration of DCs and NK cells is associated with the positive response to immunotherapy in melanoma patients [150]. In addition, NK-cell-based immunotherapy is a promising approach for treating advanced melanoma patients who are resistant to T-cell-based immunotherapy because NK cells target the HLA class-I-reduced tumour [67].

The expression of receptors on NK cells also determines its function and activity [67,152]. When activating receptors, such as DNAM-1, NKG2D (Rae1*δ* in the murine system), and NKp46, bind to their ligands CD155 and MICA/B on cancer cells, it increases NK cell cytokine production, and further promotes cancer cell apoptosis [67,68]. It has also been shown that cancer cells shed MIC, which, in turn, results in the binding of NKG2D of NK cells to the soluble MIC, causing the degradation of NKG2D and overall decreasing NK cell activity [153]. Moreover, cancer cells tend to inhibit activating receptor expression on NK cells and upregulate inhibitory receptor expression, such as TIGIT, on NK cells [67]. CD155 has been shown to be highly expressed in melanoma cells [154], and the overexpression of CD155 in cancer cells induces DNAM-1 internalised in NK cells, resulting in reduced NK cell activity [155] (Figure 2).

## 4. Processes of Resistance to Immunotherapy Responses

Unlike in targeted therapies, it is more challenging to predict treatment responses and outcomes of immune therapies solely based on several markers, such as PD-L1, CTLA-4, T-cell abundances, or phenotypes. A better understanding of the mechanisms underlying melanoma responses to immunotherapy would greatly benefit the specific therapy selection for a patient and improve overall treatment outcomes. Biomarkers, which are associated with immune responses and/or cancer cell malignant features, may be used to predict treatment outcome [156].

### 4.1. Tertiary Lymphoid Structures (TLSs)

Tertiary lymphoid structures (TLSs), which are formed by CD20^+^ B cells, CD4^+^ T cells, CD8^+^ T cells, and DCs, are lymphoid-like structures found in non-lymphoid tissue under pathological conditions, such as chronic inflammation and cancer [157,158]. TLS formation increases antigen presentation and cytokine-regulating signalling, which are associated with improvement of prognosis in cancer patients [159]. Moreover, TLSs promote immunotherapy response in melanoma [158,160,161], with high TLS formation found in those individuals that respond to immunotherapy compared to refractory patients [161]. Moreover, intratumourally injected low-dose STING in murine melanoma B16F10 tumour-bearing mice upregulates infiltrating CD8^+^ T cells and DCs that associate with TLS formation, resulting in delayed tumour growth [162]. Thus, it has been suggested that the induction of TLS formation can be a therapeutic target for improving the response to immunotherapy [157,162].

### 4.2. Therapy-Resistant Cancer Cells

In the tumour microenvironment, cancer cells tend to express and secrete proteins that trigger different mechanisms to suppress immune responses. One example is the migratory inhibition factor (MIF), which correlates with poor prognosis [163], and its receptor, CD74, found expressed on tumour-associated macrophages, DCs and MDSC [164]. MIF-CD74 signalling suppresses CD8^+^ T-cell infiltration and promotes tumour growth, which can be overcome by the blockade of MIF-CD74 signalling [164,165]. In addition, MIF also induces HIF-1α and PD-L1 expression on melanoma cells, thereby inducing further mechanisms of resistance to immunotherapies [165].

Activin-A, part of the TGF-β signalling pathway, is found to be highly expressed in tumour-associated macrophages and cancer cells in melanoma patients with metastatic disease [166]. Melanoma-derived activin-A suppresses CXCL9-CXCR3 signalling, which in turn inhibits CD8^+^ T-cell proliferation and infiltration, overall promoting tumour growth [167]. As a consequence, melanoma patients with high activin-A expression commonly fail to respond to immunotherapies [167].

### 4.3. Cytoskeleton Remodelling

A large proportion of mutations found in melanoma involve the MAPK pathway, which can promote cytoskeleton contraction and induce therapy-resistant cells [168,169]. MAPK pathway reactivation after BRAF inhibition or anti-PD-1 therapies is commonly found in therapy-resistant cancer cells [169,170]. Re-activation of MAPK signalling induces downstream events, such as Rho-associated protein kinase (ROCK) non-muscle myosin II (NMII) [171], which contributes to cytoskeleton remodelling and promotes the proliferation of the therapy-resistant cells [170]. Within the resistant tumour microenvironment, ROCK-NMII increases the abundance of CD206^+^ macrophages and FOXP3^+^ Tregs, resulting in immunosuppression [170]. Moreover, the EMT marker ZEB1 is also found to be highly expressed in the melanoma cancer cells and is associated with resistance to MAPK inhibitors [172] and immunotherapy [173], in which the latter is associated with a lower secretion of IFN-*γ* and TNF-α and induction of TGF-β2 secretion [173].

### 4.4. Immune Responses

Typically, melanoma patients with higher TILs, especially CD8^+^ T cells, generally have a better response to immunotherapies [174,175]. However, CD8^+^ T cells with higher exhausted T-cell marker abundance, such as TIGIT, are more likely to fail immunotherapies, despite high overall CD8^+^ T abundance; however, therapeutic efficacy can be improved by combining immunotherapies with TIGIT inhibitor treatments [29]. High expression of IFN-*γ* was found in patients who respond to immunotherapy [176]. IFN-*γ* upregulates MHC-I and MHC-II expression on cancer cells, which facilitates cancer cell recognition by immune cells and a subsequent induction of cytotoxic responses in the cancer cells [176,177]. Thus, the loss of MHC-I and MHC-II on melanoma cells prevents the triggering of an immune response, causing the cells to escape immune surveillance. MHC-II-positive cancer cells in melanoma patients correlate to positive anti-PD-1/PD-L1 therapeutic responses [177]. On the other hand, the expression of tumour-specific MHC-I is required for responding to anti-CTLA4 therapy [176]. Other immune cell factors, such as B-cell markers, are also important for the response to immunotherapies. Compared to non-responders in melanoma patients, B-cell-associated gene expression, such as *MZB1* and *IGLL5*, is significantly higher in therapy responders [161]. B cells have been shown to promote T-cell functions in the cancer microenvironment [161].

## 5. The Role of CD155 in the Melanoma Microenvironment and Its Potential as Immunotherapy Target

CD155 is highly expressed on melanoma cells [28] and plays a unique and emerging role in immunotherapy resistance [29]. As such, CD155 has been proposed as a potential target for immune therapy [33]; thus, understanding the role of CD155 in cancer immunoregulation might be beneficial and improve immunotherapy outcomes (Figure 3).

Two CD155 isoforms have been identified: the membrane-bound version of CD155 (mCD155) and soluble CD155 (sCD155) [183]. sCD155 is encoded by the alternative splicing isoforms CD155β and CD155γ [183], whereas mCD155 is encoded by CD155α and CD155*δ* [179]. Unlike mCD155, sCD155 lacks a transmembrane domain [179], which could result in the triggering of different functional pathways. CD155 is the ligand of three immunoglobulin-superfamily receptors: DNAM-1, TIGIT, and CD96 [27]. The main structural difference of the three receptors is the number of extracellular immunoglobulin-like domains. CD96 [184], DNAM-1 [185], and TIGIT [186] have three, two, and one extracellular immunoglobulin-like domains, respectively. Between these receptors, TIGIT has the highest binding affinity to CD155; thus, TIGIT has a dominant role in immune regulation [187,188,189]. Furthermore, these receptors are mainly expressed by NK cells, CD4^+^ T cells, and CD8^+^ T cells [190,191], and trigger both activating and inhibitory immune responses [27]. Briefly, when CD155 binds to TIGIT on cytokine-induced killer cells, which are a heterogeneous population of CD3^+^CD56^+^ NK T cells [192], it inhibits the production and secretion of cytokines, thereby suppressing cell-mediated immune responses [193]. On the contrary, when CD155 binds to DNAM-1 on NK cells, phosphorylation of the DNAM-1 intracellular domain induces LFA-1-dependent intracellular signalling, which enhances the cytokine secretion and cytotoxicity of NK cells [194]. During cancer development, immune cells lose the capacity to eliminate cancer cells, thereby contributing to cancer progression [195]. Under physiological conditions, the expression of CD155 is low, but significantly increases in cancer [154,196,197,198], which makes it a relevant possible cause for resistance to immunotherapy [29,30,31,32]. Thus, understanding immune regulation mechanisms triggered by CD155 in melanoma will provide a rational for its specific targeting to improve anti-cancer immunotherapies.

### 5.1. DNAM-1(CD226)

DNAM-1 is the only CD155 receptor that induces NK and T-cell cytotoxic activity. Upon CD155 binding to DNAM-1 on NK cells, pAkt promotes FOXO1 phosphorylation and its cytosolic translocation, resulting in NK cell activation [178]. FOXO1, a negative regulator of NK cell activation, undergoes degradation after phosphorylation [199]. The percentages of tumour-infiltrating immune cells, including CD4^+^ T cells, CD8^+^ T cells, and NK cells, are significantly reduced in DNAM-1^−/−^ mice [178], suggesting that DNAM-1 plays an essential role as activating receptor for immunosurveillance in the tumour microenvironment [200]. Furthermore, DNAM-1/CD155 interactions also promote the IL-2, IL-12, and IL-21 cytokine-induced suppression of lung metastasis in melanoma [191]. However, this metastasis-suppressive effect by inhibiting DNAM-1 only happens within three days of melanoma injections into mice, whereas, later, this inhibition is not observed, suggesting the DNAM-1-mediated NK cell activation to only be effective during early cancer growth stages [191]. In the tumour microenvironment, overexpression of CD155 suppresses DNAM-1 on NK cells and CD8^+^ T cells, resulting in NK and CD8^+^ T-cell inactivation, resulting in uninhibited cancer cell proliferation [32,179,180,197]. It has also been shown that CD155 promotes CBL-B binding to DNAM-1, which further facilitates DNAM-1 degradation in CD8^+^ T cells [32]. In melanoma patients, a low DNAM-1 CD8^+^ T-cell abundance is associated with a poor response to immunotherapy treatment [32]. Moreover, the abundance of sCD155 in the serum is also increased in cancer patients [201]. Given that sCD155 has a higher binding affinity to DNAM-1 compared to TIGIT and CD96 [179,202], sCD155 might have systemic effects, including regulation of immune responses toward circulating cancer cells or micrometastatic deposits. sCD155 is highly capable of suppressing DNAM-1 on NK cells, leading to the inhibition of degranulation and cytokine secretion, further enhancing tumour growth and metastasis [179,180]. Furthermore, there is a significant increase in DNAM-1 abundance on CD4^+^ T cells, CD8^+^ T cells, and NK cells in CD155^−/−^ mice compared to WT mice [179]. Overall, these data suggest that CD155 suppresses anti-tumour immune responses by blocking DNAM-1 via various mechanisms on both NK and CD8^+^ T cells.

### 5.2. T-Cell Immunoreceptor with Ig and ITIM Domains (TIGIT)

TIGIT, a complementary costimulatory receptor of DNAM-1, is expressed by NK cells and TILs in melanoma [28,154], capable of triggering immune suppressive responses [189]. Upon binding to CD155, TIGIT inhibits T-cell proliferation and activation, including suppressing T-cell activation markers, CD69 and CD25 [27,188,203]. In melanoma, TILs have higher TIGIT and lower DNAM-1 expression compared to peripheral blood T cells [127,154]. TIGIT has been shown to inhibit DNAM-1 expression in cytotoxic T lymphocytes and NK cells to suppress their cytotoxic activity [180], which can be reversed by TIGIT blockade [28,154]. One mechanism used by TIGIT to downregulate DNAM-1 is inhibiting DNAM-1 homodimerization, which was demonstrated in colon cancer and breast cancer models [188] and provides a potential, testable DNAM-1 inhibition mechanism in melanoma. Additionally, PD-L1 is highly co-expressed with TIGIT in T cells [154]. Moreover, IFN-*γ* has been shown to upregulate CD155 expression and PD-L1^+^/TIGIT^+^ T-cell populations, subsequently resulting in T-cell dysfunction and TIL suppression [154]. Co-inhibition of PD-L1 and TIGIT recovers cytotoxic T-cell activity and the suppression of tumour growth [154]. However, this effect is abolished by the inhibition of DNAM-1 expression, which suggests that TIGIT suppresses cytotoxic T-cell activity through a DNAM-1-dependent mechanism [154,180]. In metastatic melanoma, both TIGIT and DNAM-1 expression are downregulated in NK cells. Furthermore, the inhibition of TIGIT failed to prevent NK cell dysfunction, potentially due to the low initial expression levels of DNAM-1 in NK cells [180]. Cytokines, such as IL-15, promote DNAM-1 and TIGIT expression in NK cells through the STAT3 pathway [180]. Thus, IL-15 induction combined with TIGIT inhibition promotes NK cell degranulation and impedes tumour metastasis [180]. Taken together, the data published so far demonstrate that CD155 induces an imbalance of TIGIT and DNAM-1 expression in the tumour microenvironment, further suppressing T-cell and NK cell activity, consequently promoting tumour growth and metastasis.

### 5.3. CD96 (TACTILE)

The expression of CD96 is upregulated in melanoma [204], and the protein is also found to be expressed by NK cells and CD8^+^ T cells [182,184], capable of further tuning the activity of these cells [181,182,205,206]. CD96 prevents IL-12-induced CD8^+^ T-cell-dependent anti-tumour effects, especially by downregulating cytokine secretion [181]. Upon inhibition of CD96, the population of IFN-*γ*^+^ T cells and their cytokine secretion significantly increases, resulting in tumour suppression [181]. However, this effect is not observed in IL-12^−/−^ mice, suggesting that IL-12 is required for the CD96-mediated inhibition of anti-tumour immune responses [181]. Similar to TIGIT, CD96 has been shown to be co-expressed with PD-1 on CD8^+^ T cells [181]. Co-inhibition of PD-1 and CD96 improves CD8^+^ T-cell activity and reduces tumour growth [181]. CD96 has also been shown to directly suppress NK cell activation and limit cytokine secretion [182]. CD96 downregulates IFN-*γ*^+^ NK cells, which, in contrast, were found and recoverable in CD96-deficient mice [182], whereas IL-1β and IL-1α were upregulated in CD96^−/−^ mice [182]. Together, the effect of the absence of CD96 is a strong suppression of melanoma lung metastasis [182]. Opposite to TIGIT, CD96 mainly limits NK and CD8^+^ T-cell cytokine secretion instead of their cytotoxicity [182]. Intriguingly, and unlike DNAM-1 and TIGIT, there is a rather limited number of reports on the role of CD96 in immunoregulation.

Recently, CD155 has been identified and described as a potential novel target for immunotherapy [207]. Supporting this idea, the absence of CD155 in cancer cells decreases cell proliferation [208], invasion [209], and migration [179]. Furthermore, the inhibition of PD-L1 and CTLA4 in CD155^−/−^ mice resulted in strongly increased anti-tumour effects compared to those of PD-L1/CTLA4 blockade in WT mice [210]. Moreover, targeting CD155 by using a non-neurovirulent rhinovirus/poliovirus chimera (PVSRIPO) showed promising anti-tumour effects in a melanoma phase I clinical trial [211]. These observations highlight that the inhibition of CD155 might act additively or synergistically with the blockade of PD-L1 or CTLA4, which supports the idea that CD155 is a potential novel target for immunotherapy. So far, there are some clinical trial studies that target CD155 or its receptors in combination with current immune therapy targets (Table 1).

## 6. Overall Summary

Melanoma is a cancer type with high immunogenicity and our developing understanding of the immune regulation within the melanoma lesion will benefit the development of more effective treatments for patients. In this review, we discuss how the immune regulation involves different cell populations and how recent findings on the role of CD155 might contribute to the immune suppression in melanoma. One of the major challenges of immunotherapy is the difficulty of predicting treatment responses, especially in the case in which advanced patients failed other, previous therapies. Although a variety of contributing factors, such as TIL populations [174] and TIGIT expression on CD8^+^ T cells [154], can modulate immunotherapy outcomes, a comprehensive understanding of resistance pathways is required to overcome these limitations. Thus, future work will describe more detailed ideal combinations of existing and emerging immunotherapy targets required to achieve a sustained and complete response in melanoma patients with limited toxicities.

## Figures and Tables

**Figure 1 cancers-16-01950-f001:**
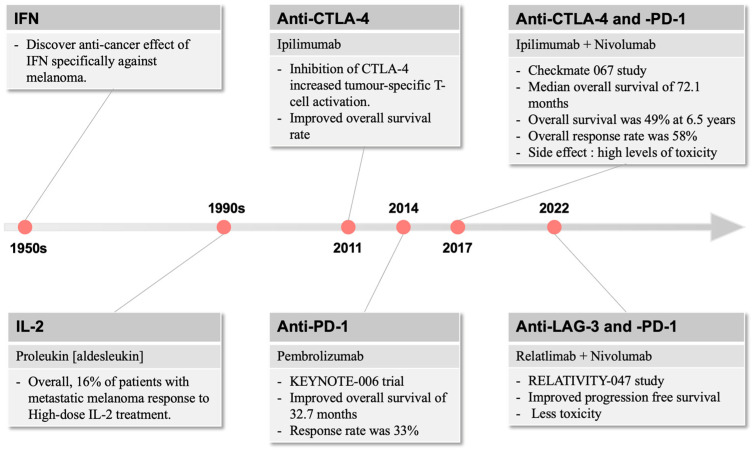
The development of immunotherapies over time. In melanoma, immune-related therapies were first described in the 1950s, which raised the interest of targeting the immune system for therapeutic use. Since 2011, immunotherapies focus on different immune checkpoint molecules, including CTLA-4 and PD-1, with more recent combination strategies providing more anti-melanoma efficacies with reduced side effect profiles.

**Figure 2 cancers-16-01950-f002:**
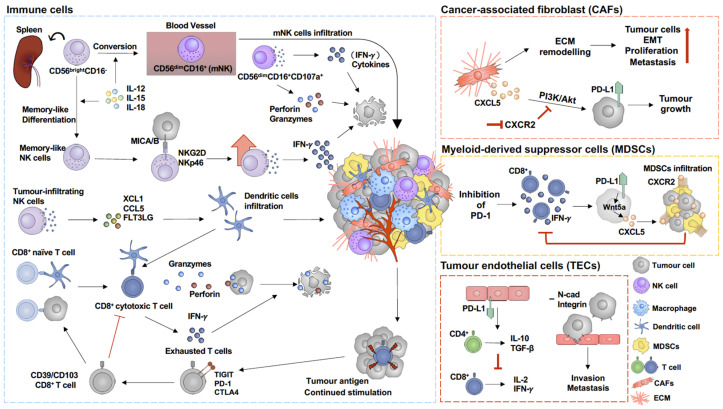
Immunoregulation in tumour microenvironment of melanoma. Activation of cancer-associated fibroblasts (CAFs) triggers extracellular matrix (ECM) remodelling, which subsequently facilitates tumour cell proliferation [59,60] and epithelial–mesenchymal transition (EMT) [61,62]. In addition, CXCL5, which is secreted by CAFs, upregulates PD-L1 expression on tumour cells through PI3K/Akt pathway activation, which promotes tumour growth and can be prevented by inhibition of CXCR2 [63]. Melanoma cells express adherent molecules, such as N-cadherin (N-cad) and integrin, which allows them to attach to tumour endothelial cells (TECs), increasing their invasion and metastasis capacity [16,64]. In addition, PD-L1 on TECs stimulates suppressive CD4^+^ T cells and upregulates IL-10 and TGF-β expression, subsequently inhibiting CD8^+^ T-cell proliferation and cytokine secretion [57]. PD-1 blockade is commonly used in immune therapy for melanoma. However, inhibition of PD-1 has been found to promote myeloid-derived suppressor cell (MDSC) infiltration into the tumour microenvironment [25,26,65]. PD-1 blockade induces CD8^+^ T-cell activation and IFN-*γ* secretion, which, in turn, upregulates Wnt5a expression in melanoma cells and further promotes CXCL5 secretion, subsequently promoting MDSC infiltration in a CXCR2/CXCL5-dependent pathway [26]. MDSC infiltration then suppresses CD8^+^ T-cell activity and IFN-*γ* secretion [26]. Activated CD107a^+^ CD56^dim^CD16^+^-mNK cells secrete perforin and granzymes and cytokines to target tumour cells in the tumour microenvironment [66]. In addition, memory-like NK cells enhance IFN-*γ* production to kill melanoma cells in an NKG2D- and NKp46-dependent manner [67]. In addition, MICA/B on tumour cells binding to NKG2D triggers NK cell cytokine production and subsequently promotes tumour cell apoptosis [67,68]. Continued stimulation by tumour antigen on CD8^+^ T cells increases exhausted T-cell populations, which are considered as dysfunctional CD8^+^ T cells [69,70]. TIGIT, PD-1, and CTLA4 are commonly expressed on exhausted T cells [71,72].

**Figure 3 cancers-16-01950-f003:**
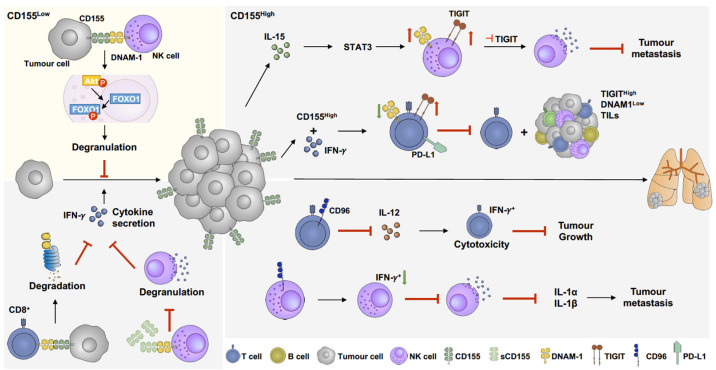
The role of CD155 in melanoma immunoregulation. CD155 manipulates immune response by activating or suppressing immune responses. When CD155 level slightly increases, the interaction between CD155 and DNAM-1 facilitates immune response by activating pAkt-FOXO1 pathway, results in NK cell degranulation and further suppression of tumour growth [178] (yellow panel). Overexpression of CD155 is commonly found in melanoma, which suppresses activation of CD8 T cells and NK cells. Overexpression of CD155 induces DNAM-1 degradation in CD8 T cells [32], which downregulates cytokine secretion and promotes tumour growth. In addition, soluble CD155 (sCD155), which has higher binding affinity to DNAM-1, inhibits NK cell degranulation and cytokine secretion [179,180]. IL-15 enhances both TIGIT and DNAM-1 expression on NK cells through STAT3 pathways. Combination of IL-15 and TIGIT inhibition facilitates NK cell degranulation and further suppresses tumour metastasis [180]. Furthermore, IFN-*γ* upregulates CD155 expression and PD-L1^+^/TIGIT^+^ CD8 T-cell populations, which suppresses DNAM-1 activation on CD8 T cells and further inhibits CD8 T-cell cytotoxicity and increases TIGIT^high^DNAM-1^low^ tumour-infiltrating lymphocytes (TILs) [154]. CD96, another immune-suppressing receptor, abolishes IL-12-induced CD8 T-cell activation and further suppresses cytokine secretion and promotes tumour growth [181]. Moreover, CD96 decreases IFN^+^ NK cell populations, inhibiting NK cell degranulation and downregulating IL-1α and IL-1β expression, resulting in metastasis promotion [182].

**Table 1 cancers-16-01950-t001:** Clinical trials targeting CD155 and its receptors.

Treatment	Target	Cancer Type	Clinical Trial
COM701	CD155	advanced solid tumour	NCT03667716 [212]
NTX-1088	CD155	advanced solid tumour	NCT05378425 [213]
Vibostolimab	TIGIT	melanoma	KEYVIBE-010 [214,215]
		advanced solid tumour	KEYVIBE-001 [216]
BMS-986207	TIGIT	endometrial cancer	NCT04570839 [217]
Tiragolumab	TIGIT	non-small-cell lung cancer	CITYSCAPE [218]

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
