# Peer review of "Immune Regulation and Immune Therapy in Melanoma: Review with Emphasis on CD155 Signalling"

_cancers, 2024, doi:10.3390/cancers16111950_

Round 1

Reviewer 1 Report

Comments and Suggestions for Authors

The manuscript by Li-Ying Wu et al. deals with an interesting topic. The authors discuss the immunoregulation involved in different cell population, in addition, how CD155 contributed to immune suppression in melanoma. Because one of the major challenges of immune therapy is hard to predict treatment outcome,  immune therapy is normally given to melanoma patients who are in advanced stage, or patients failed to response to targeted therapy.

This review helps to understand the mechanism of current immune therapy and contribute to the development of addition therapeutic targets of immune therapy in melanoma.

The review is well organized and the figures are well-edited and informative. References are relevant and up-to-date.

No similar review was published recently.

Author Response

We thank this reviewer for the time and effort in providing us with the positive feedback and felt encouraged to improve the detail and readability of the manuscript. 

Reviewer 2 Report

Comments and Suggestions for Authors

It is a very well written review displaying the interplay  between melanoma and and tumor microenviroment. It is packed with concise and uptodate information and really I enjoyed reading it. To my opinion maybe a couple of things may be missing like the role of tertiary lymphoid structures and the role of RANKL in metastasis.

A few corrections/proposistions

lines 61-63 please rephrase

line 91 please rephrase

line 204 more please offer more details on INFβ mechanism

line 211 any comments on the relation CAF RANKL?

line 291 please rephrase

lines 525 526 a bit more in depth analysis

line 592 Studies have identified a trend of increased CD96 expression

Author Response

We would like to sincerely thank reviewer 2 for the evaluation and important feedback. We have comprehensively addressed all comments as good as possible.   

Reviewer 2: 

missing like the role of tertiary lymphoid structures and the role of RANKL in metastasis.

Authors' response: 

We agree that the initial manuscript lacked any mentioning of tertiary lymphoid structures. We have now included a paragraph on those on page 10 of the manuscript. RANKL is understudies in melanoma, at least to our knowledge, and there seems to be more publications on the role of RANKL in other cancer types, including oral cancers. To maintain a stringency in this already comprehensive review, we hope the reviewer accepts that we have not included RANKL in the text. 

Reviewer 2:

lines 61-63 please rephrase

line 91 please rephrase

Authors' response: 

We have reworded these two sections accordingly. 

Reviewer 2:

line 204 more please offer more details on INFβ mechanism

Authors' response: 

We have added information on the STING-induced IFNbeta induction in tumour endothelial cells and hope that this additional data is addressing the reviewer's query.

Reviewer 2:

line 211 any comments on the relation CAF RANKL?

Authors' response: 

As above, to our knowledge and literature searches, the literature on RANKL, including CAFs, is mostly done in other cancer types. As to not dilute the focus on melanoma, we opted to focus on mechanisms described for melanoma immunotherapy regulation. 

Reviewer 2:

line 291 please rephrase

lines 525 526 a bit more in depth analysis

Authors' response: 

According to the suggestions of the reviewer, we have rephrased the section line 291 and added more details and analysis to the section lines 525ff. 

Reviewer 2:

line 592 Studies have identified a trend of increased CD96 expression

Authors' response: 

We modified the part of the text and reworded this section to increase its precision.

Reviewer 3 Report

Comments and Suggestions for Authors

The present review aims to discuss the role of CD155 in melanoma's immune regulation and the possible implications of it in immune therapy strategies. The topic sounds interesting, but there are several issues in this paper. First of all, the introduction is unnecessarily long; only in paragraph 5 do the authors write about CD155. Furthermore, they discuss it briefly and confusingly, with citations from the literature that do not add anything to current scientific knowledge. The work overall is unacceptable: an unnecessarily long introduction with many absolutely non-essential citations and a title that is misleading regarding the content. The premises for a correct, concise, and clear review are absolutely not respected. Moreover, the figures have extremely long and unclarifying captions regarding the complex dynamics related to the immuno-tumor microenvironment.

Comments on the Quality of English Language

The present review doesn't match the required quality of Standard English.

Author Response

We would like to thank the reviewer for raising several important points which we have comprehensively addressed in the updated version of the manuscript. Furthermore, we provide detailed feedback and answers to the specific points raised by the reviewer:

Reviewer 3:

The present review doesn't match the required quality of Standard English.

Authors response:

Indeed, we apologise that the use of English in parts of the initial submission was substandard. We have gone through extensive edits to the text, including and especially the abstract section, to address this concern. We are confident that the manuscript is much more readable and precise in the revised version. 

Reviewer 3:

The present review aims to discuss the role of CD155 in melanoma's immune regulation and the possible implications of it in immune therapy strategies. The topic sounds interesting, but there are several issues in this paper.

Authors response:

While this is correct, the focus of the review is to provide the reader with an understanding of the current therapeutic approaches for melanoma patients, with a focus on the use of immunotherapies, then to comprehensively explore our knowledge on the intricate interplay between the different cellular and molecular components of the immune regulation, and hence the success or failure of immunotherapies, before finally exploring exciting new discoveries in the immune regulation based on CD155. To ensure that the focus is not (only) on CD155 as the reviewer suggests, we have modified the title of the review. 

Reviewer 3:

First of all, the introduction is unnecessarily long; only in paragraph 5 do the authors write about CD155.

Authors response:

The introduction is less than one page in length and provides a brief, but comprehensive, overview of the topic. As stated above, the focus is not only on the rather limited field of CD155 and its receptors, but to provide a comprehensive, up-to-date review of immune regulation in melanoma with a clinical focus on the utilisation of this data for immunotherapies. 

Reviewer 3: 

Furthermore, they discuss it briefly and confusingly, with citations from the literature that do not add anything to current scientific knowledge.

Authors response:

The review's purpose is to comprehensively describe the current state of the scientific knowledge. Our schematic figures and the text provides a novel context how the different study results might additively provide a deeper understanding of immune regulation within the melanoma microenvironment, and roles specific cells, cytokines, CD155 and its receptors and other components have. We suggest that the review article, especially in the modified version, which incorporates more relevant references and a more precise text, thanks to the feedback of all three reviewers and editor, will achieve this goal. 

Reviewer 3:

The work overall is unacceptable: an unnecessarily long introduction with many absolutely non-essential citations and a title that is misleading regarding the content.

Authors response:

We acknowledge that the original title of the manuscript might have suggested a very narrow focus of the review article, which is why we have modified it to reflect the rather comprehensive scope, ranging from current patient management praxis over immune regulation aspects in melanoma to novel findings and developments. We have curated the referenced articles and cannot identify non-essential citations. The publications we have used are, to our knowledge, the most appropriate ones to reference the findings of others. 

Reviewer 3:

The premises for a correct, concise, and clear review are absolutely not respected.

Authors response:

With all due respect, we disagree with this notion. The current review is not meant to be a brief 'news update' but is aimed at the interested reader, either with a clinical or scientific background, to gain a deep understanding of the immune regulation in melanoma, and the implications for immunotherapies. The text is as concise as it can be, given the breadth of knowledge accumulated. Where appropriate, summary findings of hundreds of publications have been made as these were not the focus of the current review, for example detailed intracellular signalling pathways. 

Reviewer 3:

Moreover, the figures have extremely long and unclarifying captions regarding the complex dynamics related to the immuno-tumor microenvironment.

Authors response:

We have revised the figure legends and tried to reduce their length, however, without reducing the important information they contain. As the reviewer correctly states, the dynamics and interactions in the melanoma microenvironment are complex, both on cellular and molecular levels. The figures we have provided in this review are highly valuable as they are providing detailed interaction patterns in the most refined and precise manner possible. Obviously, given the complex interactions and dynamics, these figures are not brief schematics. Any kind of reductionist approach to those parts of the figure/manuscript would render the article rather useless. Therefore, we are convinced that the reviewer will agree that the figures provide a comprehensive graphical summary of the different aspects of immune regulation in the melanoma microenvironment, the goal of this comprehensive review article, and in the context of the text, benefit the readers and conveying the details of the current state of knowledge in the field.